# Preparation of Polymer-Based Nano-Assembled Particles with Fe_3_O_4_ in the Core

**DOI:** 10.3390/polym15112498

**Published:** 2023-05-29

**Authors:** Jian Wang, Wenjie Zhang, Yating Zhang, Haolin Li

**Affiliations:** The Department of Materials Engineering, Taiyuan Institute of Technology, Taiyuan 030008, China; 192073317@tit.edu.cn (W.Z.); 192073101@tit.edu.cn (Y.Z.); 192074311@tit.edu.cn (H.L.)

**Keywords:** *tert*-butyl acrylate, acrylic acid, living anionic polymerization (LAP), polymerization-induced self-assembly (PISA), organic–inorganic composite nanoparticles

## Abstract

Organic–inorganic nanocomposite particles, possessing defined morphologies, represent the next frontier in advanced materials due to their superior collective performance. In this pursuit of efficient preparation of composite nanoparticles, a series of diblock polymers polystyrene-*block*-poly(*tert*-butyl acrylate) (PS-*b*-P*t*BA) were initially synthesized using the Living Anionic Polymerization-Induced Self-Assembly (LAP PISA) technique. Subsequently, the *tert*-butyl group on the *tert*-butyl acrylate (*t*BA) monomer unit in the diblock copolymer, yielded from the LAP PISA process, was subjected to hydrolysis using trifluoroacetic acid (CF_3_COOH), transforming it into carboxyl groups. This resulted in the formation of polystyrene-*block*-poly(acrylic acid) (PS-*b*-PAA) nano-self-assembled particles of various morphologies. The pre-hydrolysis diblock copolymer PS-*b*-PtBA produced nano-self-assembled particles of irregular shapes, whereas post-hydrolysis regular spherical and worm-like nano-self-assembled particles were generated. Utilizing PS-*b*-PAA nano-self-assembled particles that containing carboxyl groups as polymer templates, Fe_3_O_4_ was integrated into the core region of the nano-self-assembled particles. This was achieved based on the complexation between the carboxyl groups on the PAA segments and the metal precursors, facilitating the successful synthesis of organic–inorganic composite nanoparticles with Fe_3_O_4_ as the core and PS as the shell. These magnetic nanoparticles hold potential applications as functional fillers in the plastic and rubber sectors.

## 1. Introduction

Poly (acrylic acid) (PAA) and polyacrylate exhibit utility as adsorbents [1], anti-scaling agents [2], dispersants [3] and thickeners [4] in modern industrial production. This versatility arises from the presence of carboxylic acid in the PAA polymer main chain, which facilitates interactions with various particles, surfaces, or small molecules via hydrogen or ionic bonding. However, the “living”/controlled polymerization of acrylic acid monomers was challenging due to the strong complexation of their carboxyl groups [5,6,7,8,9]. Typically, researchers resort to carboxyl-protected acrylate monomers, such as *tert*-butyl acrylate (*t*BA) and *tert*-butyl methacrylate (*t*BMA), for polymerization. These are subsequently subjected to deprotection reactions, yielding polymers with carboxylic acid, which are then employed in the development of diverse functional materials [10,11,12,13,14,15,16,17,18].

For example, Yoshinaga et al. [19] synthesized poly(*tert*-butyl acrylate)-*b*-poly(methyl methacrylate) (P*t*BA-*b*-PMMA) diblock copolymer by living anionic polymerization. This was then hydrolyzed to obtain poly (acrylic acid)-*b*-poly(methyl methacrylate) (PAA-*b*-PMMA) and doped with titanium dioxide (TiO_2_) nanoparticles using the sol-gel method. The resulting PAA@TiO_2_-*b*-PMMA organic–inorganic composite film material exhibited high transparency in the visible region, and the content of TiO_2_ in the composite film material was adjusted by changing the ratio of PAA segments, which in turn changed the refractive index of the film material. In a separate study, Ryu et al. [20] synthesized polystyrene-*b*-poly(2, 2, 2-propenoic acid trifluoroethyl ester) (PS-*b*-PTFEAs) from polystyrene-*b*-poly(*tert*-butyl acrylate) (PS-*b*-P*t*BAs) via a high conversion rate ester exchange reaction of *t*BA monomer units with trifluoroethanol, catalyzed by polyphosphoric acid, which can form ordered film materials. Hvilsted et al. [21] synthesized an amphiphilic diblock copolymer with a sulfhydryl group at the end, polycaprolactone-*b*-polyacrylic acid (HS-PCL-*b*-PAA), and introduced gold nanoparticles to form stable and well-dispersed organic–inorganic nanocomposite assembled particles using the complexation of the carboxyl group on the PAA segment with metal ions. Among them, PCL, which constituted the core of the nanoassemblies, demonstrated biocompatibility and exhibited high drug permeability to small drug molecules, while PAA, which constituted the shell, was biocompatible and had mucosal adhesion, and thus could be used as a drug carrier for the treatment of bladder cancer. Chen et al. developed a flexible photonic crystal using a relatively soft hydrophobic monomer *t*BA. The obtained P*t*BA photonic crystal has good flexibility, excellent hydrophobic properties, and bright tunable structural color. It was found that a wide color gamut of 112% and 10 times higher luminous intensity could be achieved by placing the P*t*BA film on the back side of a liquid crystal display [22]. Bettinger and other researchers developed a star-shaped poly (acrylic acid)-*b*-poly(*n*-butyl acrylate)/polyaniline (Star-AA_20_-*b*-BA_80_/PANI) conjugated polymer network with good flexibility and electrical conductivity for use as wearable electronics or implantable biomedical devices [23].

Polymerization-Induced Self-Assembly (PISA) was a revolutionary method for producing polymer-based nano-assembled particles [24,25,26,27,28,29]. In the PISA procedure, polymerization and self-assembly may occur concurrently, simplifying the operation steps, and the resulting nanoassemblies have a controllable morphology and high solid content (up to 50%) [30,31,32,33,34,35,36]. Living Anionic Polymerization (LAP) was characterized by relatively simple and easily controlled polymerization rate, high monomer conversion rate (close to 100%), a high activity of polymer living species, and no side reactions [37,38,39,40,41,42,43]. In our group’s previous research work, combining the controllability of LAP and the self-assembly function of PISA, the LAP PISA technique was developed and realized the LAP PISA process based on isoprene, styrene and *p*-*tert*-butylstyrene monomers, and the nano-self-assembled particles with different morphologies such as spheres, worms and vesicles were efficiently prepared [44,45]. However, the absence of post-modifiable functional groups on the prepared nano-assembled particles limits the application of the nano-assembled particles obtained by LAP PISA in practice.

In view of the versatility of poly (*tert*-butyl acrylate), we introduced *tert*-butyl acrylate functional monomers into the study of LAP PISA in order to expand the scope of LAP PISA monomers. Herein, we successfully synthesized the diblock copolymer PS-*b*-P*t*BA nano-self-assembled particles using cyclohexane as the solvent, trace Tetrahydrofuran (THF) as the polarity modifier, *n*-Butyllithium (*n*-Bu^−^Li^+^) as the initiator, 1,1-Diphenylethylene (DPE) as the active species conversion agent, styrene as the first monomer, and *t*BA as the second monomer by the LAP PISA technique. Subsequently, the *tert*-butyl group on the *t*BA monomer unit in the diblock copolymer was hydrolyzed to the carboxyl group using CF_3_COOH to obtain PS-*b*-PAA-based nano-self-assembled particles, and the resulting nano-self-assembled particles were characterized by Gel Permeation Chromatography (GPC), Nuclear Magnetic Resonance (NMR), Transmission Electron Microscopy (TEM) and Dynamic Light Scattering (DLS) to investigate the factors influencing the morphology of the nano-self-assembled particles. Finally, we also used the nano-self-assembled particles with carboxyl groups in the core as polymer templates, using the carboxyl group to complex with the metal precursors, and the organic–inorganic nanocomposite particles with a Fe_3_O_4_ core and polystyrene shell were prepared by reduction reaction (Figure 1).

## 2. Materials and Methods

### 2.1. Materials and Chemicals

Styrene (St, 99%), 1,1-Diphenylethylene (DPE, 98%) *tert*-Butyl acrylate (*t*BA, 99%), and *tert*-Butyl methacrylate (*t*BMA, 98%) were purchased from Adamas Reagent Co., Shanghai, China, stirred and dried with calcium hydride for 20 h, distilled under reduced pressure and then used. Cyclohexane (99.9%) was purchased from Adamas Reagent Co., Shanghai, China, dried with calcium hydride and then distilled at atmospheric pressure. Tetrahydrofuran (THF, Water ≤ 30 ppm (by K.F.), 99.9%, stabilized with BHT, safedry, safeseal), *n*-Butyllithium (*n*-Bu^−^Li^+^, 1.6 M solution in hexanes, safeseal), FeCl_2_ (99.5%, powder), FeCl_3_·6H_2_O (98%), and Aqueous ammonia (NH_3_·H_2_O, 25–28%, AR) were purchased from Titan Scientific Co., Ltd., Shanghai, China, and used directly. Trifluoroacetic acid (TFA, 99%), and Calcium hydride (CaH_2_, 95%, AR) were purchased from Aladdin Biochemical Technology Co., Ltd., Shanghai, China, and used directly.

### 2.2. Preparation of the Macromolecular Initiator PS

Firstly, 84.70 mL cyclohexane (66.07 g), 1.1 mL THF (1.00 g) and 7.6 mL styrene (6.93 g) were added to a dry and clean ampoule in turn, stirred and protected by nitrogen; then, *n*-Bu^−^Li^+^ (1.60 mmol/mL) was slowly added dropwise to the mixed system to remove impurities until the color of the mixture changed to golden yellow, indicating that the impurities were removed; finally, 1 mL of *n*-Bu^−^Li^+^ (1.60 mmol/mL), which was used to initiate styrene polymerization, was immediately added to the ampoule, and the reaction was stirred at 0 °C for 30 min to obtain a macromolecular initiator PS with active species at the end, and a 5 mL sample was left for characterization.

### 2.3. Conversion of the Active Centre of the Macromolecular Initiator PS

First, 5.0 mL of cyclohexane (3.90 g) and 0.8 mL of DPE (1.11 g) were added sequentially to a 25 mL dry, clean, nitrogen-protected round-bottom flask; then, *n*-Bu^−^Li^+^ (1.60 mmol/mL) was slowly added to the flask dropwise with stirring to consume impurities until the color of the mixture changed to orange; finally, the mixture was added to the ampoule of macromolecular initiator PS, the color of the system changed to deep red, and the reaction was stirred for 20 min at 0 °C to prepare the macromolecular initiator PS with DPE-Li as the active center.

### 2.4. Preparation of the Diblock Copolymer PS-b-PtBA

First, 6.2 mL of *t*BA (5.54 g) was added to 25 mL of a dry, clean round-bottom flask filled with nitrogen, stirred and DPE-Li was slowly added dropwise to consume impurities until the color of the mixture changed to pale yellow; then, *t*BA monomer was added to the ampoule after the removal of impurities to start the LAP PISA process, and the red color of the reaction system immediately faded. The system gradually became turbid and the viscosity increased as the polymerization proceeded. After stirring the reaction for 1 h at room temperature, the diblock copolymer PS-*b*-P*t*BA was obtained; finally, the reaction system was exposed to air to terminate the polymerization.

### 2.5. Preparation of PS-b-PAA Nanoself-Assembled Particles

PS-*b*-PAA was obtained by hydrolyzing the *t*BA monomer unit on the diblock copolymer PS-*b*-P*t*BA obtained by LAP PISA using CF_3_COOH. Specific experimental steps: the polymer PS-*b*-P*t*BA (7.50 g) was added to a 20 mL round bottom flask and stirring was started, 0.6 mL CF_3_COOH (0.90 g) was slowly added dropwise to the flask with a syringe and the hydrolysis reaction was carried out at room temperature for 24 h. During the hydrolysis, the system was observed to become more turbid, and finally the product PS-*b*-PAA was obtained and dispersed in cyclohexane for characterization.

### 2.6. Preparation of PS-b-PAA@Fe_3_O_4_ Nanocomposite Particles

Using the carboxyl group on the PAA chain segment to complex with the metal precursors FeCl_3_·6H_2_O and FeCl_2_, Fe^3+^ and Fe^2+^ were introduced into the core of PS-b-PAA nano-self-assembled particles, and NH_3_·H_2_O was used as the reducing agent to generate PS-*b*-PAA@Fe_3_O_4_ organic–inorganic nanocomposite particles in situ (Figure 1). The specific experimental steps are as follows. First, 0.3012 g of PS_34_-*b*-PAA_20_ diblock copolymer nano-assembled particles were dispersed in 10 mL DMF, 0.3962 g of FeCl_2_ and 0.8434 g of FeCl_3_·6H_2_O (25 equivalents of carboxyl group) were added, and stirred under nitrogen protection for 24 h; subsequently, 9 mL of NH_3_·H_2_O (in excess) was added and reacted at 50 ℃ or 30 min, and then aged at 80 °C for 1 h, followed by centrifugation at 1000 rpm for 5 min to remove the larger size of the aggregated nanoparticles to obtain PS_34_-*b*-PAA_20_@Fe_3_O_4_ nanocomposite particles.

### 2.7. Characterization Instruments

A 400 MHz Fourier transform nuclear magnetic resonance spectrometer (^1^H NMR, Bruker-AVANCE III HD, Ettlingen, Germany): the chemical shift internal standard was tetramethylsilane (TMS), and the solvents used for the test were deuterated chloroform (CDCl_3_), deuterated methanol (CD_3_OD) and deuterated dimethyl sulfoxide (DMSO-d_6_).

Gel permeation chromatography (GPC, THF, Agilent-1260, San Diego, CA, USA): The G1310B pump, G1362A differential detector, G1314F UV detector and gel column (500 Å, molecular weight detection range 500 to 4 × 10^6^ Da) were connected. The calibration standard was polystyrene (PS), the test temperature was 35 °C, and the mobile phase was chromatographic grade THF at a flow rate of 0.5 mL/min. The relative molecular weights and distributions of polymers soluble in THF were measured.

Dynamic light scattering (DLS, Malvern Zetasizer Nano ZS90, Sydney, Australia): the measurement range was from 0.3 to 5000 nm and the scattered light detection angle was 173°. The solvents used for the measurements were cyclohexane and THF. The size of the nanoparticles was measured.

Transmission electron microscopy (TEM, JEOL JEM-1230, Tokyo, Japan): accelerating voltage of 80 kV, line resolution of 0.2 nm, and point resolution of 0.36 nm. The diblock copolymer nanoassembled particles were formulated into a mixture with a concentration of 0.1~0.3% *w*/*w*. A drop of the mixture was placed on the copper web coated with carbon film and dried in air for 24 h to remove the solvent for TEM characterization. The solvent used was cyclohexane.

Thermogravimetric analyzer (TGA, Pyris 1, East Lyme, CT, USA): temperature range from room temperature to 1000 °C, sample capacity: 60 mL/1.3 g, balance sensitivity: 0.1 mg, temperature rise/fall rate range from 0.1 °C/min to 200 °C/min. The thermal decomposition temperature of the material was characterized in a nitrogen atmosphere at a rate of 10 °C/min from 50 °C to 700 °C.

X-ray Diffractometer (XRD, D/max2200PC, Tokyo, Japan): used for the measurement of the samples. Cu target Kα radiation, tube voltage of 40 kV, tube current of 30 mA, scanning speed of 5 °C/min, and scanning angle of 5 to 80°.

## 3. Results and Discussion

### 3.1. Characterization of PS-b-PtBA in LAP PISA

In the LAP PISA based on St and *t*BA, *n*-Bu^−^Li^+^ was used as the initiator, cyclohexane as the solvent, and trace amounts of THF as the polarity modifier. In the first stage of the polymerization reaction, the St monomer was first polymerized via LAP to produce the macromolecular initiator PS with reactive species at the end. In the second stage of the polymerization, DPE with large site resistance was added for the conversion of the reactive center to reduce the initiation activity of the macromolecular initiator PS, and then the *t*BA monomer was added to the system for the LAP PISA process. Due to the poor solubility of P*t*BA segments in cyclohexane, with the polymerization of *t*BA monomer, the solubility of the resulting diblock copolymer PS-*b*-P*t*BA gradually decreased, and the diblock copolymer PS-*b*-P*t*BA started to assemble to form nano self-assembled particles, thus realizing the LAP PISA process. A series of LAP PISA formulations based on PS-*b*-P*t*BA were designed by varying the ratio of molecular weight of P*t*BA and PS segments (*M_n_*_,*t*BA_/*M_n_*_,PS_), solids content, and other factors. The assembly morphology of the obtained nano self-assembled particles was investigated, and the polymerization formulations and data were summarized as shown in Table 1.

#### 3.1.1. NMR and GPC Characterization

The crude products of the polymers obtained from the two polymerization stages of the LAP PISA process were characterized using NMR. As shown in Figure 2, in the ^1^H NMR spectra of the crude product of the macromolecular initiator PS Figure 2a, 6.30–7.30 ppm was the chemical shift of H on the benzene ring (–C_6_***H***_5_) in the PS segment, and in the ^1^H NMR spectrum of the crude product of the diblock copolymer PS-*b*-P*t*BA Figure 2b, 1.50 ppm was the chemical shift of H on the *tert-butyl* (–C(C***H***_3_)_3_) in the P*t*BA segment, 2.05–2.35 ppm was the chemical shift of H in the P*t*BA chain (–OCH_2_C***H***O–), and 6.30–7.30 ppm was the chemical shift of H on the benzene ring (–C_6_***H***_5_) in the PS. Apparently, no signals of double bonds on St or *t*BA monomers were detected in the ^1^H NMR spectra, which proves that St and *t*BA monomers have been completely consumed in the polymerization process of LAP PISA. 

We performed GPC characterization of the two polymerization stages of the LAP PISA process based on the diblock copolymer PS-*b*-P*t*BA. The GPC curves of the macromolecular initiator PS prepared in the first polymerization stage showed a single peak and a narrow molecular weight distribution (*M_w_*/*M_n_* < 1.10), which proved that the macromolecular initiator PS was successfully synthesized (Figure 3a–c). The diblock copolymer PS-*b*-P*t*BA was prepared in the second polymerization stage, and the molecular weight and molecular weight distribution of the block polymer could not be characterized by GPC because the resulting PS-*b*-P*t*BA was insoluble in THF, N,N-Dimethylformamide (DMF) and other solvents. This can be attributed to the cross-linking side reaction of the *t*BA monomer during the second stage of LAP PISA.

#### 3.1.2. DLS and TEM Characterization

In order to demonstrate that the diblock copolymer PS-*b*-P*t*BA formed cross-linked nanoparticles during LAP PISA, we dispersed the diblock copolymer PS-*b*-P*t*BA nano-self-assembled particles in cyclohexane and THF (a good solvent for PS and P*t*BA blocks), respectively, for DLS characterization. The DLS results of PS-*b*-P*t*BA nano-self-assembled particles dispersed in cyclohexane are shown in Figure 4a. When the solid content was 15% w/w, the molecular weight of the macromolecular initiator PS was kept in the range of 3500~4000 g/mol, and the *M_n_*_,*t*BA_/*M_n_*_,PS_ were designed to be 0.4/1, 0.8/1, and 1.2/1 to obtain PS_35_-*b*-P*t*BA_11_,PS_34_-*b*-P*t*BA_20_ and PS_36_-*b*-P*t*BA_32_, respectively. It can be observed in the corresponding DLS curves that the average size of the diblock copolymer nano-self-assembled particles PS-*b*-P*t*BA increases with the increase in *M_n_*_,*t*BA_/*M_n_*_,PS_. The DLS results of the diblock copolymer PS-*b*-P*t*BA nanoself-assembled particles dispersed in THF are shown in Figure 4b, and the size distribution of the PS-*b*-P*t*BA nanoself-assembled particles synthesized by LAP PISA in THF was in the range of 90–1100 nm. Since both PS and P*t*BA segments have good solubility in THF, the DLS results of PS-b-PtBA nano-self-assembled particles dispersed in THF should theoretically be less than 10 nm, but the PS-*b*-P*t*BA nano-self-assembled particles prepared by LAP PISA exhibited larger size in THF, which further proved that the *t*BA monomer was cross-linked in situ in the second polymerization stage of LAP PISA and formed cross-linked stable nano-assembled particles. The morphology of the diblock copolymer PS-*b*-P*t*BA nanoassemblies obtained by LAP PISA was characterized using TEM. As shown in Figure 4c–e, the diblock copolymers PS_35_-*b*-P*t*BA_11_ and PS_34_-*b*-P*t*BA_20_ were obtained by the LAP PISA process when the solid content was 15% *w*/*w* and the designed *M_n_*_,*t*BA_/*M_n_*_,PS_ was 0.4/1 and 0.8/1, and their TEM images were shown in Figure 4c,d. The irregular morphology of the nano-assembled particles can be observed. The TEM image of the diblock copolymer PS_36_-*b*-P*t*BA_32_ obtained by the LAP PISA process with a fixed solid content of 15% *w*/*w* and increasing *M_n_*_,*t*BA_/*M_n_*_,PS_ to 1.2/1 was shown in Figure 4e, from which spherical micelles with a diameter of about 300 nm can be observed. It can be concluded that the LAP PISA system based on PS-*b*-P*t*BA was prone to the formation of irregularly shaped nano-self-assembled particles, which can be attributed to the lower glass transition temperature of the second segment P*t*BA (Tg = 40 °C) and the weaker ability of self-assembly to form the stable core [46]. We hydrolyzed the P*t*BA chain segment in PS-*b*-P*t*BA nano-self-assembled particles into the PAA segment to increase the glass transition temperature of the core block.

### 3.2. Characterization of PS-b-PAA Nanoself-Assembled Particle

The P*t*BA segments in PS-*b*-P*t*BA nano-self-assembled particles were hydrolyzed to prepare nano-self-assembled particles with a core containing carboxyl functional groups. The series of PS-*b*-PAA were prepared using CF_3_COOH hydrolysis of the *t*BA monomer unit on the diblock copolymer PS-*b*-P*t*BA obtained from LAP PISA (Table 2). 

#### 3.2.1. NMR Characterization

We purified the samples of the macromolecular initiator PS, the diblock copolymer PS-*b*-P*t*BA and the diblock copolymer PS-*b*-PAA obtained by hydrolysis and characterized them using ^1^H NMR. Figure 5a showed the ^1^H NMR spectrum of the purified treated macromolecular initiator PS, Figure 5b was the ^1^H NMR spectrum of the purified treated PS-*b*-P*t*BA, and Figure 5c was the ^1^H NMR spectra of PS-*b*-PAA after purification treatment. The area of peak ‘j’ at 2.10–2.40 ppm was used as the reference, and the peaks at 1.00–2.00 ppm in the ^1^H NMR spectra before Figure 5b and after hydrolysis Figure 5c were integrated and compared, respectively, and it was calculated that the P*t*BA segment had been completely hydrolyzed into the PAA segment, and the hydrolysis conversion rate was 100%.

#### 3.2.2. TEM and DLS Characterization

The diblock copolymers PS_35_-*b*-P*t*BA_11_,PS_34_-*b*-P*t*BA_20_ and PS_36_-*b*-P*t*BA_32_ were hydrolyzed to obtain PS_35_-*b*-PAA_11_,PS_34_-*b*-PAA_20_ and PS_36_-*b*-PAA_32_ nano-self-assembled particles containing carboxyl groups in their cores, respectively. The morphology of the hydrolyzed nano-self-assembled particles was characterized using TEM. The results were shown in Figure 6. The TEM image of PS_35_-*b*-PAA_11_, from which regular spherical micelles with a size of about 20 nm can be observed in Figure 6a; Figure 6b,b’ was the TEM image of PS_34_-*b*-PAA_20_, from which regular spherical micelles with a size of about 35 nm can also be observed. Increasing the degree of polymerization of PAA, a mixture of short worm-like micelles and spherical micelles with a size of about 50 nm can be observed in the TEM images of PS_36_-*b*-PAA_32_ (Figure 6c,c’). The TEM characterization results illustrate that when the solids content was 15% w/w and the molecular weight of the macromolecular initiator PS was kept in the range of 3500~4000 g/mol, the size of the PS-*b*-PAA diblock copolymer nanoparticles gradually becomes larger and the morphology changed from spherical micelles to short worm-like micelles as the polymerization degree of the core block PAA increased. The size of the nano-assemblies was characterized using DLS. Figure 6d shows the DLS measurements of the nano self-assembled particles PS-*b*-PAA obtained after hydrolysis. The average diameter of the nano self-assembled particles was about 20 nm in the DLS curve of the spherical micelle PS_35_-*b*-PAA_11_, about 50 nm in the DLS curve of the spherical micelle PS_34_-*b*-PAA_20_, and the average diameter of the nano-self-assembled particles was about 80 nm in the DLS curves of the mixed system of worm-like micelles and spherical micelles PS_36_-*b*-PAA_32_. These DLS characterization results showed a variation trend that was in good agreement with the TEM results. 

Based on the above characterization results, we can conclude that PS-*b*-PAA was prepared by CF_3_COOH hydrolysis of the *t*BA monomer unit on the diblock copolymer PS-*b*-P*t*BA obtained from LAP PISA, and due to the high glass transition temperature of PAA (*T*_g_ = 106 °C) [47], the diblock copolymer PS-*b*-PAA can all form regular-shaped nano-self-assembled particles. The assembly morphology of the nano-self-assembled particles obtained from PS-*b*-PAA could be adjusted by changing the degree of polymerization of PAA at a solid content of 15% *w*/*w*, and the molecular weight of the macromolecular initiator PS was kept in the range of 3500–4000 g/mol.

### 3.3. Characterization of PS-b-PAA@Fe_3_O_4_ Nanocomposite Particles

Using the complexation between the carboxyl group on the PAA segment and FeCl_3_·6H_2_O and FeCl_2_ metal precursors, Fe^3+^ and Fe^2+^ were introduced into the core region of PS-b-PAA nano-self-assembled particles, and then NH_3_·H_2_O was used as the reducing agent to generate PS-b-PAA@Fe_3_O_4_ organic–inorganic nanocomposite particles in situ. 

#### 3.3.1. TEM and DLS Characterization

Inorganic–organic nanocomposite particles PS_34_-*b*-PAA_20_@Fe_3_O_4_, obtained from the hydrolysis product PS_34_-*b*-PAA_20_ with a solid content of 15% *w*/*w* as a template, were used as an example. The morphology was characterized by TEM by dispersing the inorganic–organic nanocomposite particles into cyclohexane. Figure 7a shows the TEM image of PS_34_-*b*-PAA_20_, where spherical micelles with an average diameter of about 50 nm can be observed; the TEM image of PS_34_-*b*-PAA_20_@Fe_3_O_4_ after modification of Fe_3_O_4_ nanoparticles in the core region of the micelles is shown in Figure 7b,b’, and spherical self-assembled particles with a larger size of about 100 nm can be observed. The size of PS_34_-*b*-PAA_20_@Fe_3_O_4_ was significantly larger compared to PS_34_-*b*-PAA_20_ without Fe_3_O_4_ modification, and the PS shell layer on the surface of the spherical micelles can be observed in the magnified Figure 7b’ of PS_34_-*b*-PAA_20_@Fe_3_O_4_. The DLS characterization results are shown in Figure 7c. The DLS curves of PS_34_-*b*-PAA_20_ nano-assembled particles show an average size of about 50 nm, and the DLS curves of PS_34_-*b*-PAA_20_@Fe_3_O_4_ nano-assembled particles show an average size of about 100 nm, again demonstrating that the size of the nano self-assembled particles after the modification of Fe_3_O_4_ was larger than that before the modification, which was consistent with the TEM measurements.

#### 3.3.2. TGA Characterization

TGA was utilized to separately analyze nano-assembled particles pre- and post-Fe_3_O_4_ modification. As shown in Figure 8, the weight loss was 45.2% in the TGA curve of the organic–inorganic nanocomposite particles PS_34_-*b*-PAA_20_@Fe_3_O_4_, while the weight loss was 99.7% in the TGA curve of the diblock copolymer nanoassemblies PS_34_-*b*-PAA_20_, which was almost all thermally decomposed at 485 °C. Obviously, this was due to the introduction of Fe_3_O_4_ in polymer nanoassemblies.

#### 3.3.3. XRD Characterization

The nanoassemblies before and after modification of Fe_3_O_4_ were characterized separately using XRD. As shown in Figure 9, the characteristic peaks of Fe_3_O_4_ (2θ = 18.3°, 30.2°, 35.7°, 43.4°, 53.4°, 57.3° and 62.7°) correspond to the diffraction peaks of (111), (220), (311), (400), (422), (511) and (440) crystalline planes of cubic phase Fe_3_O_4_, respectively. The XRD spectra of PS_34_-*b*-PAA_20_ and PS_34_-*b*-PAA_20_@Fe_3_O_4_ were compared, and the same PS-*b*-PAA non-crystalline diffraction peaks were found around 20°, while the characteristic absorption peaks of Fe_3_O_4_ appeared in the XRD spectra of PS_34_-*b*-PAA_20_@Fe_3_O_4_, proving that Fe_3_O_4_ was successfully bound to the core of the polymeric nano-assembled particles.

## 4. Conclusions

In summary, utilizing the LAP PISA technique, diblock copolymer nano-self-assembled particles PS-*b*-P*t*BA were synthesized. The employed methodology incorporated cyclohexane as a solvent, *n*-Bu^−^Li^+^ as an initiator, trace tetrahydrofuran as a polarity modifier, DPE as the active species conversion agent, St as the first monomer and *t*BA as the second monomer. The nano-assembled particles were insoluble in THF, DMF and other solvents, and the DLS results in THF showed a large size, demonstrating that the *t*BA monomer undergoes cross-linking side reactions during LAP PISA to form cross-linked stable nanoparticles with irregular morphology. Subsequently, CF_3_COOH was used to hydrolyze the *tert*-butyl group on the *t*BA monomer unit in the diblock copolymer PS-*b*-P*t*BA obtained by LAP PISA into the carboxyl group, and the prepared diblock copolymer PS-*b*-PAA could self-assemble into spherical and short worm-like micelles with the core containing carboxyl functional groups. The morphology of the nano-self-assembled particles formed by PS-*b*-PAA gradually evolved from spherical micelles to a mixture of spherical micelles and short worm-like micelles as the degree of polymerization of the core block PAA increased. Finally, the nano-self-assemblies containing carboxyl groups were used as polymer templates, and the PS-*b*-PAA@Fe_3_O_4_ organic–inorganic composite nano-assembled particles were prepared by introducing Fe_3_O_4_ into the core of the polymer nanoassemblies using the complexation between the carboxyl group and the metal precursors. It is expected that PS-*b*-PAA@Fe_3_O_4_ spherical nanoparticles can be dispersed uniformly in an elastomer and used as magnetic functional fillers to prepare an electromagnetic shielding elastomer.

## Data Availability

Not applicable.

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
