# Peer review of "Preparation of Polymer-Based Nano-Assembled Particles with Fe3O4 in the Core"

_polymers, 2023, doi:10.3390/polym15112498_

Round 1

Reviewer 1 Report

The abstract contains abbreviations that are not explained, thus making the understanding of the main ideas of the study impossible.

The introduction section contains details that are more suitable for the Discussion section, as the introduction should gradually introduce the reader to the aspects of the study.

The characterization instruments should be presented after the synthesis procedures.

In the preparation of macromolecular initiator PS, it is unclear why the authors first added  n-Bu-Li+ (1.60 mmol/mL) to the mixture to remove impurities, and then the same solution was added again to initiate styrene polymerization. Why wasn't the polymerization initiated after the first addition? How are the impurities removed?

The authors did not actually demonstrate the formation of magnetite within the polymer nanoassemblies. While the TGA analysis shows a lower mass loss for the presumably Fe3O4 modified polymer particles, this doesn't actually demonstrate that it is caused by the presence of fe3o4 nanoparticles, only that there is an inorganic phase present. Thus, the authors should perform additional studies that would prove the presence of magnetite (e.g., XRD, SAED).

English must be improved, there are many grammatical errors, as well as formatting errors (e.g., Line 395 - In summary,  The).

Reviewer 2 Report

The manuscript " Preparation of polymer-based nano-assembled particles with 2 Fe3O4 in the core" is interesting to read. However, there are some suggestions to improve the manuscript.

1. Please provide information such as the full form of the LAP PISA when appears for the first time in the manuscript.

2.Write the full name of CF3COOH in the abstract

3. There are scare recent studies involved in this article. It is suggested to do compare results with recent studies.

4. The authors have written "Figure 4(e), from which 277 spherical micelles with a diameter of about 30 nm can be observed" However from the figure the size cannot be observed. There is no size mentioned for the particular particle. Please do the needful in the figure and what about the size of Figures 4 c and d?

5. The same I can see in Figure 6.

5.

The English needs to be improved.

Reviewer 3 Report

1.     Line 208, please explain why author chose this three (0.4/1, 0.8/1, 1.2/1) ratios of molecular weight of PtBA and PS segments.

2.     In Table2, please show the morphology of different samples.

3.     Line 275, “The TEM image of the diblock copolymer PS36-b-PtBA32 obtained by the LAP PISA process with a fixed solid content of 15% w/w and increasing Mn,tBA/Mn,PS to 1.2/1 was shown in Figure 4(e), from which  spherical micelles with diameter of about 30 nm can be observed.” The diameter of PS36-b-PtBA32 nanoparticles showed in Figure 4(a, b, e) didn’t match the result (from which spherical micelles with diameter of about 30 nm).

4.     In 3.2.2, the figure numbers were wrong.

5.     In figure 6. The amplification of TEM images, b’ and c’, should select from b and c, respectively, and indicated.

6.     In 3.3, why authors chose PS34-b-PAA20 to synthesize the PS-b-PAA@Fe3O4, please explain.

7.     Line 375,” the PS shell layer on the surface of the spherical micelles can be observed in the 375 magnified Figure 7(b’) of PS34-b-PAA20@Fe3O4.” Please the indicate the PS shell in the TEM image.

8.     Please measure the zeta charge of the nanoparticles, which is important as drug delivery carriers.

Some spell mistakes need to be avoided, for example, “firstiy” in line 11.

Round 2

Reviewer 1 Report

The authors have addressed all my comments. At this point, I recommend the publication of the manuscript.

English was improved.